# Exploring BenzylethoxyAryl Urea Scaffolds for Multitarget Immunomodulation Therapies

**DOI:** 10.3390/ijms24108582

**Published:** 2023-05-11

**Authors:** Raquel Gil-Edo, German Hernández-Ribelles, Santiago Royo, Natasha Thawait, Alan Serrels, Miguel Carda, Eva Falomir

**Affiliations:** 1Inorganic and Organic Chemistry Department, University Jaume I, 12071 Castellón, Spain; ragil@uji.es (R.G.-E.);; 2Curapath, Benjamin Franklin Avenue 19, 46980 Paterna, Spain; 3Institute of Agronomic Engineering for Development, Polytechnic University of Valencia, 46022 Valencia, Spain; 4Cancer Research UK Scotland Centre, Institute of Genetics and Cancer, University of Edinburgh, Crewe Road South, Edinburgh EH4 2XR, UK

**Keywords:** aryl urea, benzylethoxyaryl urea, angiogenesis, PD-L1, VEGFR-2, TLCRs, immune checkpoints, PD-1, TIM-3, LAG-3, CD11b, CD69, OX-40, CD8+ T cell, LCK, Fyn, LAT, ZAP70

## Abstract

Thirteen benzylethoxyaryl ureas have been synthesized and biologically evaluated as multitarget inhibitors of VEGFR-2 and PD-L1 proteins to overcome resistance phenomena offered by cancer. The antiproliferative activity of these molecules on several tumor cell lines (HT-29 and A549), on the endothelial cell line HMEC-1, on immune cells (Jurkat T) and on the non-tumor cell line HEK-293 has been determined. Selective indexes (SI) have been also determined and compounds bearing *p*-substituted phenyl urea unit together with a diaryl carbamate exhibited high SI values. Further studies on these selected compounds to determine their potential as small molecule immune potentiators (SMIPs) and as antitumor agents have been performed. From these studies, we have concluded that the designed ureas have good tumor antiangiogenic properties, exhibit good inhibition of CD11b expression, and regulate pathways involved in CD8 T-cell activity. These properties suggest that these compounds could be potentially useful in the development of new cancer immune treatments.

## 1. Introduction

Immune surveillance is a biological process in which the host identifies and targets cells presenting foreign antigens for destruction whilst restricting reactivity against self-antigens to avoid potentially destructive autoimmunity [1]. In the context of malignant diseases, chronic antigen stimulation in the absence of T-cell co-stimulation may lead to a state of T-cell exhaustion, allowing unchecked tumor progression. In this sense, the complex communication established between cancer and immune cells, which are part of the tumor microenvironment (TME), is highly relevant for tumor cell expansion. The dual opposing role of the immune system, that is, host protection versus tumor promotion, may lead to the inhibition or promotion of tumor growth by shaping tumor immunogenicity or by the inhibition of the protective antitumor responses, respectively [2]. These apparent paradoxical functions of the immune system reveal that cancer immune surveillance is only a part of the function of immunity, known as cancer immunoediting, which is described as a dynamic process whereby the immune system not only protects the host against cancer development but also defines the character of emerging tumors [3]. Cancer immunoediting comprises three steps: elimination of altered cells, also known as protection or cancer immunosurveillance; equilibrium of regulatory and effector immune cells; and escape that leads to tumor growth. Equilibrium phase results in the immune system’s elimination of the transformed cells as fast as they grow. On the contrary, the evasion phase induces the evasion of cancer cells to control by the immune system due to the ability of TME to hijack these effector cells [3]. Consequently, escape from immune control is now recognized to be one of the ‘Hallmarks of Cancer’ [4,5].

In this context, it is clear that communication networks between immune and cancer cells, sculpted by the immunoediting tumor microenvironment, contribute to malignant progression. Immune regulatory checkpoints, such as PD-1/PD-L1 or OX-40/OX-40L, and soluble leverage factors, such as vascular endothelial growth factor (VEGF), IL-6, TGF-β, or TNF-a perform crucial roles in these immunosuppressive networks thus causing unsatisfactory clinical responses in immunotherapy of advanced cancer. It is, therefore, imperative to not only search for new agents that can help to overcome the interaction between tumors and the immune system but also determine their impact on the immune cells within the TME [6].

Over the last decade, there has been remarkable progress in developing more effective therapies targeting tumor cells and the tumor microenvironment (TME). Among the most successful of these innovative immunotherapies are inhibitors of immune regulatory checkpoints, such as those that target the PD-1/PD-L1 axis. Pembrolizumab (Keytruda^®^), an anti-PD-1 antibody, is a clinically used antibody designed to reinvigorate cytotoxic CD8+ T-cell responses by blocking the interaction between PD-L1/2 and PD-1, enabling CD8+ T-cells cells to recognize and kill cancer cells. The concentration of PD-L1 ligand in cancer cells can be higher than 90% than in healthy ones, making it a highly targetable protein. Immunologic drugs, such as pembrolizumab, have been used to treat many types of cancer, driving, in some cases, advanced diseases into remission, although, unfortunately, they are only effective in a low proportion of treated patients. For example, success rates in the treatment of renal cell carcinoma are between 20% and 30% [7]. Therefore, challenges remain.

The overexpression of the soluble pro-angiogenic vascular endothelial growth factor A (VEGF-A) and vascular endothelial growth factor receptor-2 (VEGFR-2) activation drive abnormal angiogenesis. Anti-angiogenic therapies, based on the administration of small non-peptidyl molecules, such as sorafenib or sunitinib, have been used in clinics for two decades. Tumor angiogenesis leads to abnormal vessel formation that promotes immune evasion, and it has been recently demonstrated that the combination of anti-PD-1/PD-L1 based immunotherapy and antiangiogenic treatment by using (VEGFR-2) inhibitors results in intra-tumor immune modulation and enhances the anti-tumor efficacy of PD-1/PD-L1 blockade [6]. 

Next-generation cancer immunotherapies are designed to broaden the therapeutic repertoire by targeting alternative immune inhibitory checkpoint receptors such as lymphocyte-activation gene-3 (LAG-3) [8] and T cell immunoglobulin and mucin-domain containing-3 (TIM-3) [9] to reinvigorate T-cell responses. Contrary to immune inhibitory checkpoints, T cell co-stimulatory molecules, such as CD69 [10] or OX-40 [11], have also attracted broad interest because they are induced following TCR engagement, and they are involved in T-cell activation and inhibition of tumor progression [12].

Despite this progress, many questions about the immuno-modulatory roles of TCRs in T cell function remain unanswered and little is known about the mechanisms by which these receptors mediate their inhibitory or stimulatory actions. What is clear is that signaling pathways initiated by these are necessary for effective responses to any kind of pathogen or tumor cell. The interaction between the T-cell receptor (TCR) and major histocompatibility complex proteins (MHC) dictates these responses. Since the TCR has no enzymatic profile, a pool of tyrosine kinases is required to initiate the downstream signaling cascade leading to cytokine production and differentiation in T cells. For example, LCK and Fyn are the most proximal cytoplasmic signaling molecules to be recruited by the T-cell receptor [13]. In the case of LCK, its activation downstream of the TCR leads to phosphorylation cascades, including other tyrosine kinases, such as ZAP70 and LAT [14]. Loss or inhibition of one of these disrupts TCR signaling and blocks the T cell response. These tyrosine kinases are also promising therapeutic targets [10] and have been just considered for the development of new cancer immunotherapies. For example, Dasatinib, an oral small molecule inhibitor of Abl and Src family tyrosine kinases (SFK), with a potent ability to inhibit LCK activity, inhibits TCR-mediated signal transduction, cellular proliferation, cytokine production, and in vivo T-cell responses and, so, is considered a potential immunomodulator agent [15].

The main goal of the research we present here is the development of molecules capable of simultaneously inhibiting PD-L1 and VEGFR-2 and their effect on immune cells by assessing the expression of co-inhibitory and co-stimulatory T-cell receptors (PD-1, TIM-3, LAG-3, CD69, and Ox-40). 

Over the last five years, our research has focused on the screening of compounds capable of simultaneously blocking biological targets of special relevance, not only in the cancerous process but also in the maintenance of the tumor microenvironment (TME) [16].

By means of targeting studies on VEGF/VEGFR-2 and PD-L1, we are widening the range of new scaffolds available for drug discovery in the context of anticancer immunotherapies. The halophenyl urea unit is one of these scaffolds leading to promising small molecule immunomodulator agents due to their multitarget action [17,18,19]. For the designing of the structures, we took into account both the results obtained in our previous studies, describing the action of several sets of aryl urea derivatives U-1 and U-2 bearing a styryl moiety (see Figure 1), and the structures of small molecule PD-1/PD-L1 inhibitors, described in the literature, ones bearing a urea unit, which were developed by Aurigene, [20], and others bearing a biphenyl unit linked to a further aromatic ring through a benzyl ether bond, developed by Bristol-Meyers-Squibbs (see as an example structure of BMS-8 in Figure 1) [21]. Based on this information, we developed new derivatives, generically labeled as U-3 in Figure 1, bearing an aryl urea moiety connected to another aromatic group by a flexible chain through the intermediacy of an ether functionality. 

Previous docking studies [17] allowed us to determine that the binding sites between these proposed substrates bearing a pseudo-styryl aryl urea unit are the same as those found for the developed by Bristol-Meyers-Squibb. Thus, these compounds are introduced into a hydrophobic groove formed by the amino acids Tyr56, Met115, Ile116, Ala121, and Tyr123 and promote the dimerization of the PD-L1 protein so that the formation of the PD-1/PD-L1 complex is inhibited by a dual pathway: the inhibitors occupy part of the area involved in the PD-1/PD-L1 interaction and, at the same time, when the dimer is formed between two PD-L1 molecules, one of them has the opposite orientation to that required to interact with PD-1. Consequently, the interaction between PD-1 and PD-L1 is not possible. On the other side, the binding site in the kinase domain of VEGFR-2 also possesses a hydrophobic groove in which aromatic rings can be accommodated, and we determined that the proposed ureas mimicked the hydrogen bonds network shown by sorafenib [18,19]. Here, we present the synthesis and the biological validation of these new U-3 derivatives (see Figure 1 for specific structures), including their effect on immune cells.

## 2. Results

### 2.1. Synthesis of Aryl Urea Derivatives

The synthesis of the benzylethoxyaryl-ureas began with the preparation of 3-((4-methoxybenzyl) oxy)aniline **I.5** (see Figure 1). Thus, (4-methoxyphenyl) methanol **I.1** was converted into 1-(bromomethyl)-4-methoxybenzene **I.2** upon treatment with PBr_3_. An S_N_2 reaction on **I.2** with the phenoxide, generated in situ by reaction of nitrophenol **I.3** with K_2_CO_3_, afforded 1-((4-methoxybenzyl)oxy)-3-nitrobenzene **I.4**. which was converted into aniline **I.5** by hydrogenation. The desired ureas were synthesized by the reaction of the corresponding anilines with triphosgene, followed by the addition of compound **I.5** to the reaction mixture [22,23,24]. In our original synthetic scheme, we planned to obtain the ureas by reaction of carbamate **14**, prepared by the reaction of benzyl chloroformate **I.6** with aniline itself, with a set of substituted anilines. However, the reaction of carbamate **14** with different anilines under various reaction conditions afforded the desired ureas in very low yields. Nevertheless, we decided to biologically evaluate carbamate **14** as a means of comparison with ureas **1**–**13**, even though it was not used for the synthesis of these. 

### 2.2. Biological Evaluation

#### 2.2.1. Cell Proliferation Inhibition

The ability of all the synthesized compounds to affect cell viability was studied by MTT assay using the human tumor cell lines HT-29 (colon adenocarcinoma), A-549 (pulmonary adenocarcinoma), and towards HEK-293 (human embryonic kidney cells), which does not express PD-L1 [25], human microvessel endothelial cells (HMEC-1) and Jurkat T cells. This assay allowed us to establish the corresponding IC_50_ values (expressed as the concentration, in μM, at which 50% of cell viability is achieved), shown in Table 1. Table 1 also includes IC_50_ values for the reference compounds sorafenib and BMS-8.

None of the synthetic compounds had any effect on the proliferation of immune Jurkat T-cell, or endothelial cells HMEC-1. On the other hand, most of the compounds showed antiproliferative activity in the micromolar range on the tested cancer cell lines HT-29 and A-549, comparable to that shown by reference compounds sorafenib and BMS-8. It is observed that compounds **3** (*m*-fluorophenyl urea), **4** (*o*-fluorophenyl urea), **12** (*m*-methoxyphenyl urea), and **13** (*o*-methoxyphenyl urea) had no effect on cancer cell proliferation. In contrast, **6** (*m*-clorophenyl urea) was only active in HT-29. As regards activity on HEK-293, it should be noted that carbamate **14** and *p*-aryl urea-type compounds, **2** (*p*-fluorophenyl urea), **5** (*p*-clorophenyl urea), **8** (*p*-bromophenyl urea) and **11** (*p*-methoxyphenyl urea), exhibited IC_50_ values above 100 μM while the rest of the compounds exhibited inhibitory action at low micromolar dose (see Table 1). 

In the case of carbamate and *p*-aryl urea-type compounds, the inhibitory effect on cell proliferation was significantly higher on both cancer cell lines, HT-29 and A-549, than on the others (HEK-293, and endothelial and immune cells). This allowed us to determine their selectivity for cancer celinesine against endothelial or immune owhich what could translate into in vivo very low toxicity. Moreover, the fact that HEK-293 lacks both studied targets allowed us to establish which compounds were the most selective, so we determined a selectivity index (SI, see Table 2) by dividing the IC_50_ mean against HEK-293 by the IC_50_ mean against both studied tumor cell lines. For the rest of the derivatives, the selectivity indexes were below 1, which means poor selectivity towards cancer cells in their inhibitory effect. For this reason, compounds with higher selectivity indexes, shown in Table 2, are the ones that were selected for further biological studies. 

#### 2.2.2. Effect on Cellular PD-L1 and VEGFR-2 in Cancer Cell Lines

To assess the effect of the selected compounds on the biological targets PD-L1 and VEFGR-2, we performed the study on both cancer cell lines HT-29 and A-549 and by flow cytometry technique. Both membrane and total PD-L1 and VEGFR-2 were relatively determined using DMSO-treated cells as a negative control and BMS-8 and sorafenib as reference compounds. For these assays, cells were incubated for 24 h in the presence of the corresponding compounds at 20 µM concentration. No significant effect on membrane PD-L1 or VEGFR-2 was achieved for either of the compounds. Table 3 shows the effects exhibited by the selected compounds on total PD-L1 and total VEGFR-2, in both cell lines.

In general, compounds were more active in HT-29 than in A-549, on which no significant effect was found. As far as the action on HT-29, compound **8** (*p*-bromophenyl urea) was the most active one, with inhibition rates around 60% for both PD-L1 and VEGFR-2 in HT-29, carbamate **14** inhibited around 40% of PD-L1 and 60% of VEGFR-2 and compound **2** (*p*-fluorophenyl urea) showed inhibition rates around 50% on both targets. In contrast, **11** (*p*-methoxyphenyl urea) had no inhibitory effect.

In view of the good results obtained for these selected compounds, we decided to extend the study to a third cancer cell line, MCF-7 (Human breast cancer adenocarcinoma). First, we determined the IC_50_ values of the derivatives and their SI indexes (see Table 4). Then, we determined the effect of the selected compounds on the biological targets, PD-L1 and VEGFR-2, by flow cytometry. Again, DMSO-treated cells were used as a negative control, and BMS-8 and sorafenib as reference compounds. As usually, the assay was performed at a concentration of 20 µM for 24 h. The results in Table 4 show the effects exhibited by the selected compounds on total and membrane PD-L1 and VEGFR-2 in MCF-7 cell lines. 

In this case, we found very selective compounds towards cancer cells and some significant results in both membrane and total, PD-L1 and VEGFR-2. Interestingly, compound **11** (*p*-methoxyphenyl urea), which had no effect on the HT-29 cell line, shows around a 40% inhibition rate for total PD-L1 in the breast cancer cell line. 

Due to the mild results obtained for this cancer cell line, we determined that these compounds are more active against HT-29, so this was the selected cell line for further studies. 

#### 2.2.3. Effect of Microtube Formation on Endothelial Cells

The capacity to inhibit the formation of a new vasculature network formed by HMEC-1 was evaluated on compounds **8** and **14** at different concentrations. Table 5 shows the minimum concentration at which these compounds are active and begin to inhibit the microtube formation.

Pictures for the inhibition of neovascularization achieved by compounds **8** and **14** are displayed in Figure 2.

A comparison of the minimum active concentration values to IC_50_ values for the HMEC-1 cell line (see Table 1) shows that the compounds exhibited antiangiogenic action while they have no effect on endothelial cell proliferation.

#### 2.2.4. Effect on Cancer Cell Viability in Co-Cultures with Monocytes THP-1

We also studied the effect of the selected compounds on tumor cell proliferation in the presence of human monocytic leukemia cell line THP-1. This assay was carried out using HT-29 as the cancer cell line and different proportions of immune cells. Standard proportions for this assay were 1:5 cancer/immune cells. Additionally, we also carried out this assay using a 2:1 proportion of cancer cells as regards immune THP-1 cells. Assays were performed after 24 and 48 h of treatment, using 20 μM doses of selected compounds and BMS-8 as the reference compound.

The results in Table 6 show that the effect on cancer cell viability was higher after 24 h of treatment and did not depend on the proportion of cancer and immune cells. Compounds **2** (*p*-fluorophenyl urea) and **11** (*p*-methoxyphenyl urea) were the most active ones showing 20% of inhibition of cell viability both at (1:5) and (2:1) proportions after 24 h. Additionally, compound **11** was also active after 48 h of treatment with an excess of cancer-related to immune cells.

Figure 3 shows the morphological changes suffered by HT-29 cells after 48 h of co-cultured with THP-1 at (1:5) proportion. We observed that control cells preserved a morphology related to the epithelial nature of HT-29 cells while BMS-8 treated cells retained this epithelial nature, resulting in an increased and brighter cytosolic granulation. Treatment of HT-29 co-cultures with carbamate **14** lead to a slight loss of cell-to-cell contact and the appearance of clustered cells with irregular surfaces. Finally, treatment with the urea derivatives **2**, **8,** and **11** lead to less confluent cultures with a few elongated fibroblast-like appearance cells (see arrows in Figure 3) and less adhesive rounded morphology that led to cell scattering with apoptotic features. All of these are morphological changes related to the loss of their epithelial appearance by the synergic action of the compounds and immune cells.

#### 2.2.5. Effect on Immune Cell Viability in Co-Cultures of HT-29/THP-1

From the previous study, we also determined the effect of the selected compounds on the human monocyte cells THP-1 in the described co-cultures. We also assessed the effect of the compounds on the cell viability of monocultured THP-1. Table 7 shows the results obtained as the proportion of living cells after each treatment compared to non-treated ones (control). In general, none of the compounds had a significant effect on immune cell viability, neither in monoculture nor in co-cultures. Only carbamate **14** and ureas **8** (*p*-bromophenyl urea) and **11** (*p*-methoxyphenyl urea) exhibited around 15% of inhibition of immune cell viability after 24 h of treatment in excess of cancer cells (see fifth column in Table 7).

It has been demonstrated that cancer, and all the therapies associated with this illness, promote functional alterations in monocytes, such as the acquisition of immunosuppressive activity in the tumor microenvironment, which is related to the expression of CD11b an integrin that, when binds to CD18 promotes the acceleration of the invasiveness and metastasis of cancer cells [26]. Reducing the expression of CD11b has become a promising target for immune modulation in anti-cancer therapies. For that reason, we decided to study the effect of our compounds on CD11b in THP-1 cells co-cultured with HT-29. We also determine the relative amount of CD80, a common surface marker for monocytes. This study was performed on the collected cell culture from each well and gating living THP-1 cell population. 

Table 8 shows the results we obtained when we determined the percentage of CD80 and CD11b expression in membrane THP-1-related non-treated cells when they were co-cultured with HT-29 and the corresponding treatment. 

It can be observed that neither of the compounds have any effect on CD80 expression in any of the tested conditions, although a mild effect could be seen on the expression of CD11b. This effect was higher in the presence of an excess of cancer cells and after 48 h of treatment, similar to the one observed for the reference compound BMS-8. In this sense, **2** (*p*-fluorophenyl urea), **11** (*p*-methoxyphenyl urea), and carbamate **14** reduced about 20% of CD11b. In the case of **14**, the same effect was observed in all tested conditions.

#### 2.2.6. Ex Vivo Study of the Effect on Cytotoxic CD8+ T Cells (OT-1)

Cytotoxic CD8+ T cells (CTL) perform a critical role in controlling cancer development and are the most powerful effectors in the anticancer immune response. CTLs exhaustion is a dysfunctional state that leads to the loss of their anti-cancer effectiveness. For this reason, we studied the effect of our selected compounds on OT-1 cells at 20 μM for 24 h of treatment.

First, we determined by flow cytometry the relative expression of a range of T-cell inhibitory checkpoint receptors, including LAG-3, TIM-3, and PD-1, together with the T cell co-stimulatory receptors CD69 and OX-40 (Figure 4). Interestingly, three of the four compounds promoted the expression of at least some activation-induced co-inhibitory and co-stimulatory receptors, perhaps implying a heightened state of activation in response to antigen stimulation in the presence of drug treatment. 

We next studied the effect of our compounds on the signaling cascade downstream of the TCR. Figure 5 shows the effect of the compounds on the most proximal molecules to the TCR (Quantification is shown in Appendix A). Notably, none of the compounds influenced the expression of Fyn. However, both the expression and phosphorylation of LCK were increased in response to all compounds. The effects on ZAP70 and LAT were more complex, with opposing effects on total protein expression versus phosphorylation. Thus, all compounds modulated signaling downstream of the TCR in a manner that is likely to impact T-cell activation and activity. 

## 3. Discussion

We have synthesized thirteen benzylethoxyaryl ureas and one benzylethoxyaryl carbamate to determine their capability as potential multitarget inhibitors of VEGFR-2 and PD-L1 proteins to overcome resistance phenomena offered by cancer. 

In terms of their antiproliferative activity, the majority of the compounds were found to be selective towards cancer cells as their IC_50_ values and, what it is the same, their inhibitory effect on cell proliferation was significantly higher on the tested cancer cell lines HT-29, A-549, and MCF-7 than on non-tumor ones HEK-293 and endothelial and immune cells.

From the observations provided, it can be concluded that there is a relationship between the structure of the synthetic compounds and their antiproliferative activity because the carbamate and the *p*-substituted phenyl ureas are more active against cancer cells and exhibited the lowest IC_50_ values for cancer cell lines and the highest selectivity indexes. Indeed, based on these indexes, we selected four compounds for further biological studies: *p*-aryl urea-type compounds **2** (*p*-fluorophenyl urea), **8** (*p*-bromophenyl urea), **11** (*p*-methoxyphenyl urea), and carbamate **14**. 

From these further studies, we found that some of the selected compounds exhibited significant inhibitory effects on both PD-L1 and VEGFR-2 in cancer cell lines. In general, the compounds exhibited a more effective action in HT-29 than in A-549 or MCF-7, and compound **8** (*p*-bromophenyl urea) and carbamate **14** were the most active in HT-29 with almost 40% of inhibition on both targets. Moreover, both **8** and **14** exhibited good antiangiogenic properties as they both inhibited the formation of new microvessels on matrigel HMEC-1 cell cultures. 

The selected compounds were also tested for their effect on cancer cell proliferation and immune cell viability in co-culture experiments using HT-29 and THP-1 cells. Ureas **2** and **11** were found to be the most active ones in inhibiting cancer cell viability with a very mild effect on immune cell viability. Furthermore, the compounds were tested for their effect on CD11b and CD80 expression in THP-1 cells co-cultured with HT-29. While none of the compounds affected CD80 expression, several of them, including ureas **2** and **11** and carbamate **14**, were found to significantly reduce CD11b expression, which is a promising target for immune modulation in anti-cancer therapies.

Finally, the compounds were studied ex vivo to assess effects on the expression of T-cell co-inhibitory and co-stimulatory receptors, including LAG-3, TIM-3, PD-1, CD69, and OX-40, in cytotoxic CD8+ T cells (OT-1). Interestingly, most of the compounds promoted upregulation of both activation-induced inhibitory and stimulatory checkpoints, perhaps implying a state of enhanced activation. Indeed, upregulation of CD69 and OX-40 may enhance T cell activation and proliferation, both of which can be beneficial in cancer immunotherapy. Therefore, the synthetic compounds’ ability to upregulate CD69 and OX-40 in OT-1 cells is another promising finding. Moreover, although an elevated TIM-3 production could be associated with an exhausted immune system, an upregulation of TIM-3 in the absence of the inhibitory ligand may also be indicative of enhanced activation. Looking in further detail at the signaling cascade downstream of the TCR, we identified modulation of total and phosphorylated LCK, ZAP70, and LAT, further suggesting that these compounds are likely to impact signaling pathways critical for T-cell activation and function. 

Collectively, these data suggest that the designed compounds have very good tumor selectivity indexes, and antiangiogenic properties, exhibit good inhibition of CD11b expression, and regulate pathways involved in CD8+ T-cell activity. These properties suggest that these compounds could be potentially useful in the development of new cancer immune treatments.

## 4. Materials and Methods

### 4.1. Chemistry

#### 4.1.1. General Procedures

^1^H and ^13^C NMR spectra were measured at 25 °C. The signals of the deuterated solvent (CDCl_3_, DMSO_d6_) were taken as the reference. Multiplicity assignments of ^13^C signals were made by means of the DEPT pulse sequence. Complete signal assignments in ^1^H and ^13^C NMR spectra were made with the aid of 2D homo- and hetero-nuclear pulse sequences (COSY, HSQC, HMBC). High-resolution mass spectra were recorded using electrospray ionization–mass spectrometry (ESI–MS). IR data were measured with oily films on NaCl plates (oils) and were given only for relevant functional groups (C=O, NH). Experiments that required an inert atmosphere were carried out under dry N_2_ in flame-dried glassware. Commercially available reagents were used as received.

#### 4.1.2. Experimental Procedure for the Synthesis of Ureas **1**–**13**

A solution of the corresponding aniline (1.0 mmol) dissolved in dry THF (5.0 mL) was slowly dripped into a stirred solution of triphosgene (303 mg, 1.0 mmol) in dry THF (5.0 mL). Then, Et_3_N (279 µL, 2.0 mmol) was then added slowly to the reaction mixture. The resulting mixture was stirred at room temperature for 1 h, then was refluxed for 5 h under nitrogen atmosphere. After this, the reaction mixture was cooled to room temperature, and the solid was filtered off. After evaporation of the solvent in vacuo, the residue was taken up in dry THF (5.0 mL), and a THF solution (5.0 mL) of 3-((4-methoxybenzyl)oxy)aniline (**I.5**) (229 mg, 1.0 mmol) was directly added followed by Et_3_N (139 µL, 1.0 mmol) addition. The resulting mixture was refluxed under a nitrogen atmosphere overnight. Then, the solvent was removed in vacuo, and AcOEt (20 mL) was added. The organic phase was washed with aqueous HCl 10% (2 × 20 mL) and brine, and dried over Na_2_SO_4_. Then, the solvent was removed in vacuo, and the residue was recrystallized from acetonitrile and dried under vacuum to give ureas **1–13** as white solids (41–99%).

### 4.2. Biological Studies

#### 4.2.1. Cell Culture

Cell culture media were purchased from Gibco (Grand Island, NY, USA). Fetal bovine serum (FBS) was obtained from Harlan-Seralab (Belton, UK). Supplements and other chemicals not listed in this section were obtained from Sigma Chemical Co. (St. Louis, MO, USA). Plastics for cell culture were supplied by Thermo Scientific BioLite (Waltham, MA, USA). For tube formation assay, an IBIDI μ-slide angiogenesis (IBIDI, Martinsried, Germany) was used. All tested compounds were dissolved in DMSO at a concentration of 10 mM and stored at −20 °C until use.

HT-29, MCF-7, A549, HEK-293, and Jurkat cell lines were maintained in Dulbecco’s modified Eagle’s medium (DMEM) containing glucose (1 g/L), glutamine (2 mM), penicillin (50 μg/mL), streptomycin (50 μg/mL), and amphotericin B (1.25 μg/mL), supplemented with 10% FBS. The HMEC-1 cell line was maintained in Dulbecco’s modified Eagle’s medium (DMEM)/Low glucose containing glutamine (2 mM), penicillin (50 μg/mL), streptomycin (50 μg/mL), and amphotericin B (1.25 μg/mL), supplemented with 10% FBS. For the development of the tube formation assays in Matrigel, HMEC-1 cells were cultured in EGM-2MV Medium supplemented with EGM-2MV SingleQuots.

To carry out the ex vivo experiments, CD8+ cell line was maintained in Iscove’s Modified Dulbecco’s media (IMDM) supplemented with 10% FBS, L-glutamine (10 μL/mL), Penicillin/streptomycin (10 μL/mL) and β-Mercaptoethanol (50 μM).

#### 4.2.2. Cell Proliferation Assay

In 96-well plates, 5 × 10^3^ (HT-29, MCF-7, A549, HMEC-1, Jurkat, and HEK-293) cells per well were incubated with serial dilutions of the tested compounds in a total volume of 100 μL of their respective growth media. The 3-(4,5-dimethylthiazol-2-yl)-2,5-diphenyltetrazolium bromide (MTT; Sigma Chemical Co.) dye reduction assay in 96-well microplates was used. After 2 days of incubation (37 °C, 5% CO_2_ in a humid atmosphere), 10 μL of MTT (5 mg/mL in phosphate-buffered saline, PBS) was added to each well, and the plate was incubated for a further 3 h (37 °C). After that, the supernatant was discarded and replaced by 100 µL of DMSO to dissolve formazan crystals. The absorbance was then read at 550 nm by spectrophotometry. For all concentrations of compound, cell viability was expressed as the percentage of the ratio between the mean absorbance of treated cells and the mean absorbance of untreated cells. Three independent experiments were performed, and the IC_50_ values (i.e., concentration half inhibiting cell proliferation) were graphically determined using GraphPad Prism 4 software.

#### 4.2.3. PD-L1 and VEGFR-2 Relative Quantification by Flow Cytometry

To study the effect of the compounds on every biological target in cancer cell lines, the compounds were used at 20 µM dose. 

For the assay, 10^5^ cells per well were incubated for 24 h with the corresponding dose of the tested compound in a total volume of 500 μL of their growth media. 

On the one hand, to detect total PD-L1 and VEGFR-2, after the cell treatments, they were collected and fixed with 4% in PBS paraformaldehyde. After fixation, a treatment with 0.5% in PBS TritonTM X-100 was performed, and finally, cells were stained with FITC Mouse monoclonal Anti-Human VEGFR-2 (ab184903) and Alexa Fluor^®^ 647 Rabbit monoclonal Anti-PD-L1 (ab215251).

On the other hand, to detect membrane PD-L1 and VEGFR-2, the process was the same, avoiding the treatment with 0.5% in PBS TritonTM X-100 step. 

#### 4.2.4. Tube Formation Inhibition Assay

Wells of an IBIDI μ-slide angiogenesis (IBIDI, Martinsried, Germany) were coated with 15 μL of Matrigel^®^ (10 mg/mL, BD Biosciences) at 4 °C. After gelatinization at 37 °C for 30 min, HMEC-1 cells were seeded at 2 × 10^4^ cells/well in 25 μL of culture medium on top of the Matrigel and were incubated 30 min at 37 °C while are attached. Then, compounds were added and dissolved in 25 μL of culture medium, and after 24 h of incubation at 37 °C, tube formation was evaluated.

#### 4.2.5. Cell Viability Evaluation in Monoculture and in Co-Cultures

First, to study the effect of the compounds on the cell viability in monoculture, 10^5^ of cancer cell line per well were seeded and incubated for 24 h; then, the corresponding compound at 20 μM or DMSO for the positive control were added. After 24 h of incubation, cancer cells were collected with trypsin and fixed with 4% in PBS paraformaldehyde and counted by flow cytometry.

On the other hand, to study the effect of the compounds on the cell viability in co-culture with THP-1 cells, 10^5^ or 2 × 10^5^ of the HT-29 cells line per well were seeded and incubated for 24 h, then the medium was changed by one cell culture medium supplemented with IFN-γ (10 ng/mL; human, Invitrogen^®^, Waltham, MA, USA) and containing 5 × 10^5^ or 105, respectively, of THP-1 per well and the corresponding compound at 20 μM or DMSO for the positive control. After 24 h/48 h of incubation, supernatants were collected to determine THP-1 living cells. Additionally, stain cancer cells were collected with trypsin. Both types of suspension cells were fixed with 4% in PBS paraformaldehyde and counted by flow cytometry.

#### 4.2.6. CD11b and CD80 Relative Quantification by Flow Cytometry in Co-Cultures

To study the effect of the compounds on every biological target in co-cultured THP-1 immune cells with cancer cell line HT-29, the compounds were incubated for 24 h/48 h as described before. 

To detect membrane CD11b and CD80, after the cell incubation, suspended THP-1 was collected from the cell culture of each sample well, fixed with 4% in PBS paraformaldehyde, and stained with FITC Mouse monoclonal Anti-Human CD80 (Sigma-Aldrich, S. Louis, MO, USA, SAB4700142) and Alexa Fluor^®^ 647 Rabbit monoclonal Anti-CD11b (Merck, S. Louis, MO, USA, #MABF366). 

#### 4.2.7. Generation of CTL Cells from OT-1 Mice

CD8+ cells were obtained from the inguinal, axillary, brachial, and mesenteric lymph nodes previously collected from OT-1 mice. Lymph node cells were isolated by mashing through a 70 μM cell strainer. Cells were resuspended in IMDM cell media at a concentration of 0.3 million cells/mL. N4 ovalbumin peptide was added at a concentration of 10 nM, and cells were incubated for 48 h at 5% CO_2_, 37 °C. On day 2 from recollection, cells were pelleted at 1300 rpm for 5 min and washed with PBS to remove N4 peptide. The cell pellet was resuspended in fresh IMDM media in a new flask in a 0.3 million cells/mL and supplemented with IL-2 at 20 ng/mL. After incubation for a further 48 h (day 4), cells were re-seeded in fresh IMDM media at a concentration of 0.3 million cells/mL supplemented with IL-2 at 20 ng/mL. Cells were cultured for a further 48 h prior to use on day 6 in functional assays. 

#### 4.2.8. Ex Vivo Study of CTLs Exhaustion Receptors by Flow Cytometry

After the generation of CTLs, 5 × 10^5^ of these cells were seeded into individual flasks in a final volume of 2.5 mL; to this, compounds were added in a final concentration of 20 μM. After 24 h of incubation, the resulting single-cell suspension was washed with PBS twice. The resulting cell pellet was re-suspended in PBS containing Zombie NIR viability dye [1:2000 dilution (BioLegend, San Diego, CA, USA)] and incubated at 4 °C for 15 min, then pelleted by centrifugation at 1300 rpm for 5 min at 4 °C. Cells were washed with FACS buffer and pelleted by centrifugation twice. Cell pellets were resuspended in 100 μL of Fc block [1:200 dilution of Fc antibody (eBioscience, MA, USA) in FACS buffer] and incubated for 15 min. Then, 100 μL of antibody mixture [diluted in FACS buffer (antibody details listed in Appendix A)] was added to each well, and the samples were incubated for 20 min in the dark. The cells were then pelleted by centrifugation at 1300 rpm for 5 min at 4 °C and washed twice with FACS buffer as above. Finally, cells were re-suspended in FACS buffer and analyzed using a BD Fortessa. Data analysis was performed using FlowJo software. Statistics and graphs were calculated using Prism (GraphPad).

#### 4.2.9. Immunoblotting

To study the effect of the compounds in the signaling pathway of CTLs, the compound was incubated for 24 h at 20 μM. After this time, cells were collected and washed twice in PBS and lysed in radioimmunoprecipitation assay (RIPA) lysis buffer (50 mM tris-HCl at pH 7.4, 150 mM sodium chloride, 5 mM EGTA, 0.1% SDS, 1% NP-40, and 1% deoxycholate) supplemented with a protease and phosphatase inhibitor cocktail (mini complete ULTRA Protease tablet and phosSTOP tablet from Roche, Basel, Switzerland). Lysates were clarified by high-speed centrifugation (17,000 rpm for 15 min at 4 °C). Protein concentration was measured using a Micro BCA Protein Assay (Thermo Scientific, Waltham, MA, USA), and 10 μg to 30 μg of total protein were supplemented with 2X SDS sample buffer [tris (pH 6.8), 20% glycerol, 5% SDS, b-mercaptoethanol, and bromophenol blue) and boiled at 95 °C for 5 min. Samples were separated by polyacrylamide gel electrophoresis using 4% to 15% Mini-PROTEANTGX gels (Bio-Rad, Hercules, CA, USA), proteins transferred to nitrocellulose, blocked [5% bovine serum albumin (BSA) in PBS–Tween 20 (BSA/PBS-T)], and probed with either anti–glyceraldehyde-3-phosphate dehydrogenase (GAPDH; Cell Signaling Technology), anti-tubulin (Cell Signaling Technology, Danvers, MA, USA), anti–LCK (Biolegend), anti–phospho-LCK (Y394; Biolegend), anti-phospho-LCK (Y505; Cell Signaling Technology), anti-phospho-FYN (Cell Signaling Technology), anti-ZAP70 (Cell Signaling Technology), anti-phospho-ZAP70 (Cell Signaling Technology), anti-LAT (Cell Signaling Technology), or anti-phospho-LAT (Cell Signaling Technology). Bound Ab was detected by incubation with anti-rabbit and anti-mouse horseradish peroxidase secondary Ab (Cell Signaling Technology) and visualized using the Bio-Rad ChemiDoc MP Imaging System.

## Data Availability

Not applicable.

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
