# Peer review of "Exploring BenzylethoxyAryl Urea Scaffolds for Multitarget Immunomodulation Therapies"

_ijms, 2023, doi:10.3390/ijms24108582_

Round 1

Reviewer 1 Report

In this paper by Gil-Edo et al, the authors screen compounds to discover novel compounds capable of inhibiting the VEGFR and PDL1 axis. They identify several compounds with anti-proliferative properties. However, the specificity of these compounds is questionable, and it is not entirely clear how they are working mechanistically based on the data provided. Are these compounds physically interacting with VEGFR and or PDL1 to prevent downstream signaling or indirectly affecting their expression levels by other mechanisms? Also, There are some methodological flaws in the downstream validation assays performed. Specific critiques are highlighted below:

Please be more specific about the tumor types that the percentages are referring to in the intro as response rates will be quite different based on tumor types, I believe the PD1 and VEGF blockade data is referring to renal cell carcinoma?

I would not consider HEK293 cells as a “normal cells” as these cells are immortalized and highly proliferative, and can even be tumorigenic in certain cases pending on the passage number. Can state that the compounds were chosen based on their selectivity in cancer cell lines compared to HEK293 cells. Do HEK293 cells lack the putative targets of these compounds, VEGR2 and PDL1? If so, then the approach that the authors use the filter their compounds to more specific candidates makes more sense, if not, it may be flawed.

Table 3 -please change “expression” to “surface level”. Please indicate that this was by flow cytometry. Also, in the methods, it indicates a incredibly high concentration was used (100uM) for surface expression experiments which far exceeds the IC50 values for them based on the data presented in Table 1, so the loss of surface expression data may be nonspecific. As such, Was a viability dye use in these experiments to gate out dead cells? Surface expression should only be determined on live cells.

For Figure 3, These pictures are overall poor resolution for demonstrating the observations described in the text. The cells without THP-1 cells should also be shown to evaluate whether or not the THP-1 cells are responsible for the observation of elongated cells made. These changes are quite subtle and it is difficult to appreciate the significance of these findings. 

For Table7, what units is this table in? Are they in viability? If so, why are there numbers that are over 100%? You can’t have a viability over 100%....The effect on CD11b is very modest and of unclear biological significance.

For Table8, how is the surface level of the markers being determined in the mixed cultures?? Is THP-1 cells being gated out based on their FSC/SSC properties or a different marker? More details need to be provided here…..

For Figure 4, the absolute percentage expression should be shown, rather than the fold change.

For Figure 5, These western results require quantification with 3 repeat experiments with mean +/- SD. Also, the details of the length of treatment and concentration of compounds should be provided in the figure legend.  

English language if fine, only some edits required. 

Reviewer 2 Report

1. HEK 293 are indeed a cancer cell line with low tumorigenicity
2. Incubaion time should be added to the caption of Table 1
3. It is not correct to calculate selectivity index against HEK 293 cell line. Of the cell lines used, HMEC-1 are a much more correct control, as they are a truly normal cell line.
4. Table 3 and 4: expression detection method should be indicated
5. Table 3 and 4: statistical evaluation of the data should be ad
6. Table 5. The data on microtubule formation should be shown and statistically evaluated
7. Table 6, 7,8. Incubation time and statistical ealuation should be added
8. Figure 4. Detection method should be indicated
9. Figure 5. Western blots should be quantified and statistically evaluated
10. A QSAR modelling could be very interesting to detect important properties of the tested molecules

1. English puctuation requires correction

2. In some cases, the word choice seems not very appropriate

Round 2

Reviewer 1 Report

This is a resubmission of a manuscript by Gil-Edo et al. The authors have a addressed the majority of the critiques. Just a couple of minor issues that remains:

The axis of Figure 4 is labeled "free protein". Please revise this to "surface protein"

For figure 5, the authors replied that "the scale of the changes observed is so clear that quantification is not required". However, the reason the quantification is important is to evaluate if the total amount of the protein difference between control and compounds can explain the difference in the amount of phosphorylated protein. For example, if LCK and p-LCK go down by a similar amount, then it is likely that the compound is producing an effect on the total amount of the protein itself rather than on the phosphorylation event, as currently being claimed in the manuscript for LCK. Please quantify the blots and indicate the ratio of the difference of the phosphorylated to total level of the indicated proteins to provide firmer support for the claims in the manuscript.

Moderate editing of English is required.

Author Response

We have modified Figure 4 and 5 according the suggestions of reviewer.

In reference to the western blot quantification we have added these tables in supporting information document.

Reviewer 2 Report

Table 3 and 4: I suggest adding asterisks to indicate the significant differences.

The data in all of the tables are compared using a one-sample t-test, which is not correct for the multiple comparisons situation. An ANOVA with some post-test should be used instead.

I suggest using some grammar checking software like Grammarly to fix minor problems with articles and word usage.

Author Response

We have performed the changes in to Table 3 and 4 and we have realised of a mistake naming the statistical test. We have corrected it thanks to the reviewr comments.